# A Tumour and Liver Automatic Segmentation (ATLAS) Dataset on Contrast-Enhanced Magnetic Resonance Imaging for Hepatocellular Carcinoma

Félix Quinton [1,*], Romain Popoff [1,2,3], Benoît Presles [1], Sarah Leclerc [1], Fabrice Meriaudeau [1], Guillaume Nodari [2], Olivier Lopez [2], Julie Pellegrinelli [2,3], Olivier Chevallier [4], Dominique Ginhac [5], Jean-Marc Vrigneaud [1,2] and Jean-Louis Alberini [1,2,4]

1 Institut de Chimie Moléculaire de l'Université de Bourgogne, ICMUB UMR CNRS 6302, Université Bourgogne Franche-Comté, 21000 Dijon, France
2 Service de Médecine Nucléaire, Centre Georges-François Leclerc, 21000 Dijon, France
3 Service de Radiologie, Centre Georges-François Leclerc, 21000 Dijon, France
4 Service de Radiologie et Imagerie Medicale Diagnostique et Therapeutique, Centre Hospitalier Universitaire, 21000 Dijon, France
5 Laboratoire Imagerie et Vision Artificielle (ImViA), Universite de Bourgogne Franche-Comté, 21000 Dijon, France
* Correspondence: felix.quinton@u-bourgogne.fr

**Abstract:** Liver cancer is the sixth most common cancer in the world and the fourth leading cause of cancer mortality. In unresectable liver cancers, especially hepatocellular carcinoma (HCC), transarterial radioembolisation (TARE) can be considered for treatment. TARE treatment involves a contrast-enhanced magnetic resonance imaging (CE-MRI) exam performed beforehand to delineate the liver and tumour(s) in order to perform dosimetry calculation. Due to the significant amount of time and expertise required to perform the delineation process, there is a strong need for automation. Unfortunately, the lack of publicly available CE-MRI datasets with liver tumour annotations has hindered the development of fully automatic solutions for liver and tumour segmentation. The "Tumour and Liver Automatic Segmentation" (ATLAS) dataset that we present consists of 90 liver-focused CE-MRI covering the entire liver of 90 patients with unresectable HCC, along with 90 liver and liver tumour segmentation masks. To the best of our knowledge, the ATLAS dataset is the first public dataset providing CE-MRI of HCC with annotations. The public availability of this dataset should greatly facilitate the development of automated tools designed to optimise the delineation process, which is essential for treatment planning in liver cancer patients.

**Dataset:** The dataset is accessible via the following link: https://atlas-challenge.u-bourgogne.fr.

**Dataset License:** License under CC-BY-NC-SA agreement

**Keywords:** MRI; liver; liver tumours; TARE; hepatocellular carcinoma

## 1. Introduction

In addition to being the sixth most common cancer in the world, with 841,080 new cases in 2018 and estimates of over 1,000,000 new cases recorded per year by 2025, liver cancer is also the fourth leading cause of cancer mortality [1], leading to 780,000 related deaths in 2018 [2]. In particular, hepatocellular carcinoma (HCC) is the most common type of primary liver cancer in adults, representing approximately 80% of liver cancer cases.

HCC tumours can be treated with various strategies depending on the tumour staging [3]. Liver transplantation and tumour ablation (hepatectomy, radiofrequency ablation, cryoablation, etc.) [3] are the main options for early-stage HCC, while transarterial therapies such

as transarterial chemoembolisation (TACE) or transarterial radioembolisation (TARE), and systemic/targeted therapies are considered for intermediate/advanced stages of HCC. TARE consists of injecting microspheres marked with $^{90}Y$, a $\beta$-emitter isotope, selectively into the hepatic arteries vascularising the tumours, in order to irradiate and kill tumour cells. Since off-target injection of the radioactive agent into healthy tissue can lead to complications [4], the tumour must first be localised and its volume estimated for dosimetry calculation [5]. For this purpose, several imaging exams can be performed, including a contrast-enhanced magnetic resonance imaging (CE-MRI) T1-weighted exam. In clinical practice, this CE-MRI exam is used to manually delineate the liver and the intra-hepatic tumour.

The delineation process is time-consuming, requires a trained radiologist, and is subject to inter- and intra-operator variability. Therefore, automatic segmentation tools have grown in popularity in recent years. Nonetheless, access to large amounts of high-quality medical imaging data for the development and validation of these tools is difficult, mainly due to data privacy regulations and ethical issues.

Some datasets containing MRI of patients with liver tumours are already publicly available [6,7] but do not include annotations. Other datasets with liver tumour annotations have been described and used in the literature but are not publicly available. Xiao et al. [8] presented a liver tumour segmentation solution using radiomic features from T2 delay-phase CE-MRI on a dataset of 200 patients. Zhao et al. [9] proposed liver tumour detection based on generative adversarial networks, and Kim et al. [10] similarly performed region-of-interest detection of HCC on a multi-centre CE-MRI dataset. Zhao et al. [11] contributed to liver tumour segmentation on multi-modal non-contrast MRI, while Zheng et al. [12] recently took advantage of multi-phase dynamic CE-MRI to perform segmentation. Regarding other modalities, a few liver datasets with tumour annotations on CT scan data are publicly available [13–15]. Recent studies such as [16–19] also describe liver tumour segmentation solutions and results on private datasets.

The proposed "Tumour and Liver Automatic Segmentation" (ATLAS) dataset is a dataset that provides systematic annotations of liver and liver tumours on CE-MRI scans. To the best of our knowledge, the ATLAS dataset is the first publicly available dataset that provides systematic annotations of liver and liver tumours on HCC CE-MRI. The ATLAS dataset is available as part of the ATLAS challenge organised in conjunction with the second Resource-Efficient Medical Image Analysis (REMIA2) [20] at the Medical Image Computing and Computer Assisted Intervention (MICCAI) [21] workshop being held in 2023 in Vancouver, BC, Canada. The challenge and dataset are hosted at https://atlas-challenge.u-bourgogne.fr, accessed on 29 March 2023. Table 1 references the main information detailed in this paper.

**Table 1.** Dataset specifications.

| Specification Type | |
|---|---|
| Subject Area | Biomedical Imaging, Oncology |
| More specific subject area | CE-MRI segmentation of liver and HCC |
| Type of data | 3D T1 CE-MRI and associated annotations |
| How data were acquired | MR acquisitions, Siemens and GE Healthcare |
| Data format | NIfTI |
| Experimental factors | Gadolinium-based contrast agent injection |
| Experiment features | Clinical thorax/abdomen CE-MRI of HCC patients |
| Main data source location | University Hospital, Dijon 21000, France |
| Data accessibility | https://atlas-challenge.u-bourgogne.fr |

## 2. Material and Methods

### 2.1. Ethics Approval

The ATLAS dataset was largely acquired at the University Hospital, Dijon 21000, France and consists of actual clinical acquisitions of HCC. The ATLAS dataset was anonymised and processed in accordance with the rules established by the Ethical Committee of the University Hospital of Dijon. All administrative information included in the metadata has been removed, making them untraceable. Thus, in accordance with French law, it was not necessary to go through the process of obtaining an ethical approval number.

### 2.2. Data Description

The ATLAS dataset consists of 90 liver-focused CE-MRI acquisitions from 90 patients with unresectable HCC along with 90 liver and liver tumour segmentation masks. The image resolution, contrast phase, MRI sequence and MRI manufacturer are also provided for each acquisition. The cohort is composed of patients with unresectable HCC who underwent TARE from October 2012 to February 2023. The inclusion criteria were as follows: clinical indication for TARE decided by the multidisciplinary tumour committee, a CE-MRI within eight weeks prior to treatment and lesions that could be unequivocally segmented on the CE-MRI T1-weighted scan. Patients for whom a different acquisition modality was used (e.g., T2-weighted MRI, CT) were excluded.

The ATLAS dataset will be released in two parts: the first part, which is the training dataset for the ATLAS challenge, will be available from April 2023, and the second part, which is the testing dataset for the ATLAS challenge, will be downloadable from 2025. These two parts will be, respectively, referred to as the training set and the testing set in the remainder of the paper. The training set consists of the 60 oldest patients and the testing set of the 30 most recent patients.

### 2.3. CE-MRI Acquisition

An exam consists of multiple CE-MRI T1-weighted acquisitions performed with various delays after injection of a gadolinium-based contrast agent. A standard pre-treatment MRI exam comprises three phases of acquisition: late arterial, portal venous and delayed T1-weighted acquisitions at 25–30 s, 65–70 s and 180–300 s after contrast agent injection. These three phases allow us to study the tumour vascularisation. The radiologist in charge of the exam can also decide to carry out additional acquisitions in the early arterial phase at 15–25 s or in the late portal phase, i.e., 90 s after contrast agent injection.

Blood flow in HCC typically passes through the lesion and into the venous outflow faster than in a healthy liver, leading to a peak in tumour enhancement in the late arterial phase followed by a characteristic "washout" that allows us to observe a tumour hyposignal relative to the liver at the portal venous and delayed phases (Figure 1).

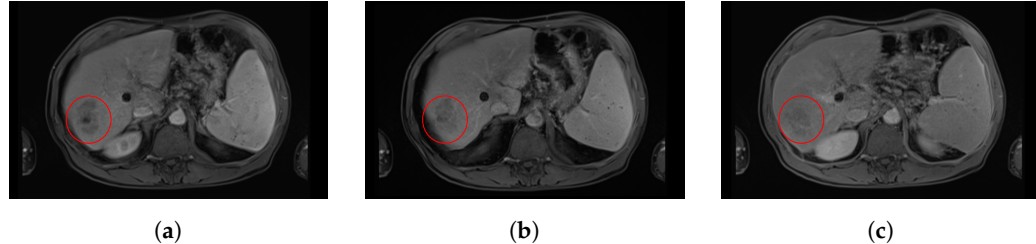

(**a**)          (**b**)          (**c**)

**Figure 1.** Evolution of tumour enhancement post-contrast agent injection in an axial slice of a contrast-enhanced magnetic resonance imaging (CE-MRI) from the same patient. This figure shows the evolution over time of the tumour enhancement in the red circle. (**a**) Axial CE-MRI slice at arterial time. (**b**) Axial CE-MRI slice at portal time. (**c**) Axial CE-MRI slice at delayed time (**c**). One may notice the enhancement in the tumour in arterial time acquisition, slowly vanishing in portal time and delayed acquisitions.

The CE-MRI exams were mainly performed on five Siemens MRI machines of 1.5 and 3T. A few exams were done on General Electric (GE) MRI 1.5T machines (Table 2).

**Table 2.** Number of acquisitions per MR equipment in the training and testing sets.

| MR Equipment | Number of Train Acquisitions | Number of Test Acquisitions |
|---|---|---|
| SIEMENS MAGNETOM SKYRA 3T | 13 | 27 |
| SIEMENS MAGNETOM AERA 1.5T | 27 | 1 |
| SIEMENS TRIOTIM 3T | 9 | 0 |
| SIEMENS MAGNETOM SOLA 1.5T | 4 | 1 |
| SIEMENS MAGNETOM AMIRA 1.5T | 0 | 1 |
| GE SIGNA HDXT 1.5T | 2 | 0 |
| GE SIGNA EXPLORER 1.5T | 2 | 0 |
| GE OPTIMA MR450W 1.5T | 1 | 0 |
| GE OPTIMA MR360 1.5T | 1 | 0 |
| GE SIGNA VOYAGER 1.5T | 1 | 0 |

Each acquisition is T1-weighted in the transversal axis with an ultrafast gradient echo sequence. On Siemens equipment, a Volumetric Interpolated Breath-Hold Examination (VIBE) sequence was used or a derived VIBE sequence, such as VIBE TWIST or VIBE CAIPIRINHA. On GE equipment, Liver Acquisition with Volume Acceleration (LAVA) or LAVA FLEX sequences were used (Table 3). Fat saturation (FATSAT) was applied for all acquisitions. Acquisitions were done with repetition time TR = 3.09–6.78 ms , echo time TE = 1.07–4.19 ms and flip angle $\alpha$ = 9.0–13.0°.

**Table 3.** Number of acquisitions per sequence in the training and testing sets.

| Sequence | Number of Train Acquisitions | Number of Test Acquisitions |
|---|---|---|
| VIBE | 24 | 15 |
| VIBE CAIPIRINHA | 22 | 2 |
| VIBE TWIST | 7 | 13 |
| LAVA | 5 | 0 |
| LAVA FLEX | 2 | 0 |

By default, on most MR equipment, four main corrections were applied to the data during image reconstruction:

- A normalisation filter that corrected the signal inhomogeneities in depth;
- An outlier removal filter applied in the frequency domain;
- 2D and 3D distortion corrections;
- Bias field correction.

Each exam resulted in three to five CE-MR images; one image was selected to be added to the dataset (Table 4). The selected image was usually one of the three post-contrast injection phases (arterial, portal or delayed) that is also used by the experts to delineate the liver and tumours. In the ATLAS dataset, early arterial images are classified as arterial phase images and late portal images as portal phase images. In some rare cases, the selected image is obtained without agent contrast injection. Finally, a CE-MRI of the ATLAS dataset consists of a 3D image of 44 to 136 transversal slices of the thorax and abdomen covering the entire liver and tumour. Each slice has a pixel spacing between 0.68 × 0.68 mm$^2$ and 1.41 × 1.41 mm$^2$, and a slice thickness between 2 mm and 4 mm.

**Table 4.** Number of patients in the different phases in the training and testing sets. The acquisition phase of the image used for delineation was not specified for seven patients who underwent examinations with contrast injection.

| Delay after Contrast Agent Injection | Number of Train Acquisitions | Number of Test Acquisitions |
|---|---|---|
| Arterial phase | 33 | 14 |
| Portal venous phase | 10 | 15 |
| Delayed phase | 8 | 1 |
| No contrast agent | 2 | 0 |
| Unknown | 7 | 0 |

*2.4. Annotation of the CE-MR Images*

Once one image was retained per patient, the liver and tumour contours including necrosis were manually delineated by an experienced MRI radiologist (more than three years of experience) on transverse slices. To estimate the inter-operator variability on the dataset, a second set of contours were manually delineated by another expert on 48 images [22]. The liver and tumour exhibited a root mean square (RMS) of the coefficient of variation below 4.5% and 6.2%, respectively. The coefficient of variation $C_{i,p}$ for a specific volume of interest $i$ ($i \in \{$liver, tumour$\}$) on a patient $p$ is defined as

$$C_{i,p} = \frac{V_{i,p}^{sd}}{V_{i,p}^{m}}, \tag{1}$$

with $V_{i,p}^{m}$ the average volume of interest in mm$^3$ segmented by two different operators and $V_{i,p}^{sd}$ the standard deviation of the two volumes. Thus, the RMS on a specific volume of interest $i$ is

$$RMS_i = \sqrt{\frac{\sum_{p=1}^{N} C_{i,p}^2}{N}}, \tag{2}$$

with the volume of interest $i$ and $N$ the number of patients.

The annotator used delineation tools available on the MIM SurePlan LiverY90 software [23] to segment both the liver and tumours. Brushes and pencil tools were used indistinctly. For the liver segmentation, the whole liver was considered; extra-hepatic arteries and gallbladder were excluded. For the tumour segmentation, all tumours were considered (if more than one was present). In practice, the expert in charge of the manual segmentation separates the tumour and necrosis because the volume occupied by necrosis should not be taken into account in the value of the tumour burden used for dosimetry calculation. However, as only a few cases showed necrosis, we decided not to include the necrosis label in the dataset. Thus, necrosis has been classified as tumour if located within the tumour and as liver otherwise. Figure 2 shows necrosis within the tumour.

After delineation, the liver and tumour contours are converted into a label image. A label is assigned to each voxel of the image depending on its location: 0 for the background, 1 for the healthy liver and 2 for the tumour.

Once the contouring process was finished on the whole dataset, CE-MRI and labelled images were, respectively, extracted under DICOM and DICOM RTStruct formats and converted into NIfTI format using CLITK [24] tools.

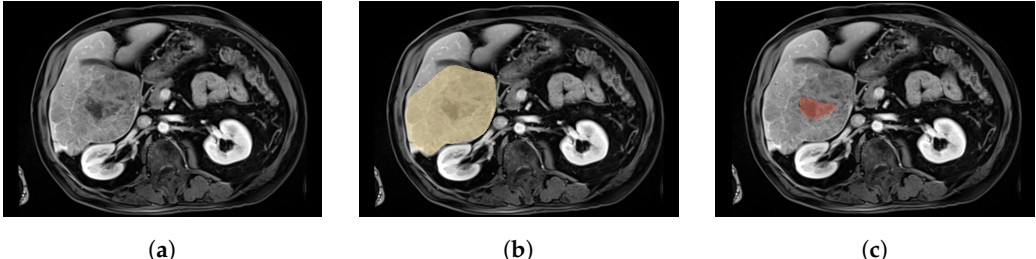

(**a**)              (**b**)              (**c**)

**Figure 2.** Axial slice of a CE-MRI from a patient with necrosis within the tumour. (**a**) Axial slice of a CE-MRI from patient with necrosis. (**b**) Axial slice of a CE-MRI from patient with tumour labels. (**c**) Axial slice of a CE-MRI from patient with necrosis labels. On (**b**,**c**), respectively representing the tumour in yellow and the necrosis in red, necrosis is located in the centre of the tumour. In this case, necrosis is therefore classified as tumour in our dataset.

## 3. Discussion

### 3.1. Intensity Distribution and Tumour Specifications

The distribution of pixel intensities per image is highly dependent on the sequence of acquisition (Table 3), so the average intensity of a given area in the ATLAS dataset may differ from one image to another. Moreover, surrounding regions such as the arms may appear on the MRI image, as in Figure 3c, resulting in higher-intensity pixels for this region compared to the abdominal organs.

HCC tumours can take various forms, as illustrated in Figure 3. The majority of the studied tumours take relatively convex forms, such as in Figure 3a. However, some cases may take heterogeneous forms with an atypical pattern (Figure 3c). Most of the time, these tumours are slightly harder to detect visually and therefore harder to delineate manually. In such cases, decreased accuracy for the label image may be assumed. Moreover, the infiltrating nature of HCC [25] may lead to multiple tumour components (Figure 3b). In addition, the tumour size varies widely across the dataset, with a maximum ratio of 850 between the largest and smallest tumours, corresponding to volumes of 2,188,369 mm$^3$ and 2574 mm$^3$, respectively. The average tumour volume in the dataset is 341,442 ± 453,942 mm$^3$. In comparison, the ratio between the largest and smallest livers is only 5, with a liver size (healthy liver + unhealthy liver) that varies from 817,954 mm$^3$ to 4,140,926 mm$^3$ and an average volume of 1,988,183 ± 655,839 mm$^3$. Thus, the tumour burden (TB = tumour volume/total liver volume) varies from 0.1% to 57.0%.

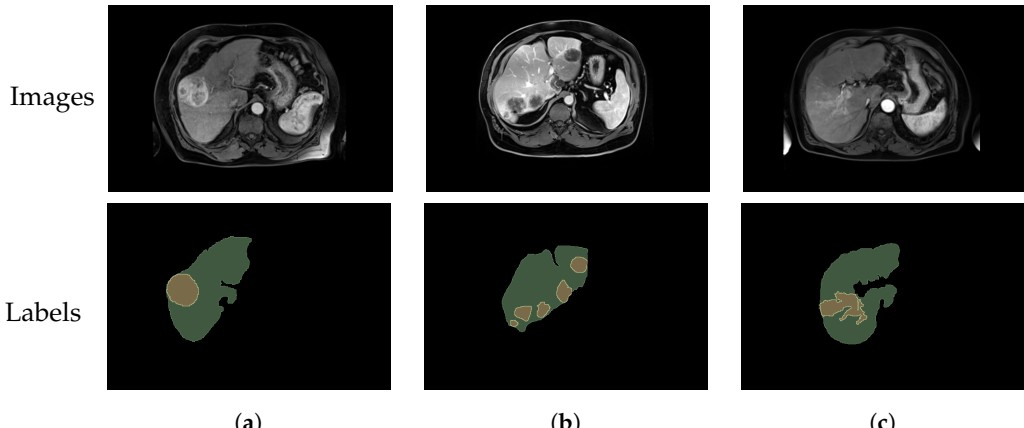

Images

Labels

(**a**)              (**b**)              (**c**)

**Figure 3.** Axial slices of CE-MRI showing different tumour shapes. (**a**) Axial slice of a CE-MRI from a patient with liver and simple tumour labels. (**b**) Axial slice of a CE-MRI from a patient with liver and multiple tumour labels. (**c**) Axial slice of a CE-MRI from a patient with liver and complex tumour labels.

Since the dataset is composed of real clinical cases, the acquisition conditions may vary. Although enhancement of the tumour is expected after injection of the contrast agent, in practice, some cases, such as in Figure 4, still present a low contrast between the tumour and the liver. This situation complicates greatly the manual segmentation by the radiologist and thus the quality of the labels.

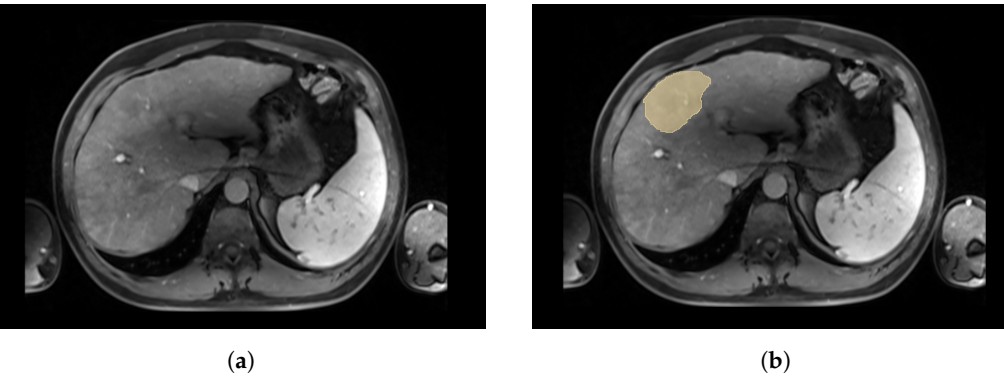

(**a**)             (**b**)

**Figure 4.** Axial slices of a CE-MRI from a patient with low contrast between tumour and liver. (**a**) Axial slice of a CE-MRI from patient with low-contrast tumour. (**b**) Axial slice of a CE-MRI from patient with tumour label. A very low contrast between the liver and the tumour in yellow was observed in the arterial phase for this patient.

### 3.2. Training and Testing Sets' Distribution

The distribution of the images between the training and testing sets is based on the acquisition date. This criterion was chosen in the context of the ATLAS challenge and aims to focus on the performance of the segmentation algorithms on future images. Future acquisitions are more likely to resemble recent acquisitions than older ones, often acquired on older machines with lower image quality and different sequences (Table 3). Thus, a domain shift is expected between the training and testing sets. For example, the average tumour size in the training set is 1.5 times larger than in the testing set, with an average volume of $386,034 \pm 483,907$ mm$^3$ in training compared to $252,258 \pm 371,385$ mm$^3$ in testing. Similarly, the number of distinct tumours per patient is higher in the training set, with 50% of patients having more than one tumour and an average of 3.1 tumours per patient, compared to 27% of patients with more than one tumour and an average of 1.7 tumours per patient in the testing set. These differences may be explained by the fact that tumours have been detected at earlier stages in recent years, resulting in smaller tumour sizes and fewer tumours per patient. It is also worth noting that only 37% of the images in the train set were acquired using 3T machines, compared to 90% for the test set. As 3T machines are known to produce higher-resolution images with a better signal-to-noise ratio, images from the test set may have higher delineation quality.

### 4. Conclusions

We introduce the first dataset of CE-MRI for HCC. The "Tumour and Liver Automatic Segmentation" dataset contains volumes from 90 patients and annotations of both liver and HCC tumour contours. The accurate annotations available in this dataset should enable researchers to lead precise experiments and help with the design and evaluation of automatic segmentation solutions. Access to labelled medical imaging datasets remains a huge challenge for the community, which hinders in particular the development of deep-learning-based solutions as they require large annotated datasets to reach efficient performance. In this context, the ATLAS dataset is the first contribution to publicly available and fully annotated MR images of liver cancer.

**Author Contributions:** Conceptualisation, J.-L.A., F.M. and B.P.; data curation, R.P., G.N., O.L., J.P., O.C. and J.-M.V.; writing—original draft preparation, F.Q.; writing—review and editing, F.Q., R.P., B.P., S.L., F.M., D.G., J.-M.V. and J.-L.A.; supervision, B.P., S.L., J.-L.A., F.M., J.-M.V. and R.P.; project administration, B.P.; funding acquisition, B.P. All authors have read and agreed to the published version of the manuscript.

**Funding:** This research was funded by Agence Nationale de la Recherche, grant number ANR-21-CE45-0002.

**Institutional Review Board Statement:** Not applicable.

**Informed Consent Statement:** Not applicable.

**Data Availability Statement:** A publicly available dataset was described in detail in this study. The presented data can be found at https://atlas-challenge.u-bourgogne.fr, accessed on 29 March 2023.

**Conflicts of Interest:** The authors declare no conflicts of interest.

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
