# Peer review of "A Tumour and Liver Automatic Segmentation (ATLAS) Dataset on Contrast-Enhanced Magnetic Resonance Imaging for Hepatocellular Carcinoma"

_data, 2012_

Round 1
Reviewer 1 Report
This paper describes a soon to be released “A Tumor and Liver Automatic Segmentation" (ATLAS) dataset that consists of contrast enhanced magnetic resonance images obtained from 90 patients with unresectable hepatocellular carcinoma encompassing the entire liver, along with manual labels for liver and liver tumor segmentation. ATLAS dataset is a first contribution to publicly available and fully annotated MR images of liver cancer. The paper is well written and the dataset it describes may constitute an important resource for future development of AI/ML techniques for MR liver tumor segmentation. I have a few comments listed below for the author's consideration to improve the manuscript.
* I assumed all 90 patients were already scanned between 10/2012 and 02/2023. While the 60 training dataset will be available from 04/2023, why the 30 testing dataset will be available almost two years later in 2025?
* Page 4, section 2.3, the unit of TR and TE should be “ms” instead of “ms-1”.
* Page 4, section 2.3, … four main corrections were applied to the data during image reconstruction (not acquisition).
* Page 4, section 2.3, I wondered if the authors would consider to release all three (arterial, portal, and delayed) phases of images to test the variability of an automated method for the whole liver and tumor segmentation on the same patient.
* Page 4, Table 4, were those 7 “unknown” scans was acquired with or without contrast agent?
* Page 4, equation 1, please provide more detail about “V”; is it the 3D volume measurement in mm3? If so, I don’t see it’s meaningful to compute the mean and standard deviation from only two numbers. A better and common way of measuring inter- and intra-observer variability is using the Dice similarity coefficient.
* Page 5, “No other pre-processing is applied”. This sentence is redundant. Unless you have done pre-processing earlier in the contouring process.
* Page 5, section 3.1, please clarify what “some intra-image normalisation strategies” were applied to your dataset. Was this part of the four main corrections that were applied to the data during an acquisition? (i.e. “A normalization filter that corrects the signal inhomogeneities in depth.”) Or there were other processing steps performed to normalize the signal intensity for all images in different phases, or for different slices in a given volume?
* Page 5, section 3.1, “… the tumour size varies a lot in the dataset with a volume from 2574 mm3 to 2 188 369 mm3 with an average volume of 341 442mm3 ± 453 942mm3.” Please double check these numbers. The largest tumor size is over 2 liters but the max TB is only 57%, which means the max liver could be almost 4 liters (true?). Please also include the mean, SD, min, and max for the whole liver volume.
* Page 6, section 3.2, the 30 testing dataset were primarily (90%) scanned from a 3T scanner, while the majority (78%) of the 60 training dataset were acquired from various 1.5T scanners. Please discuss potential performance discrepancy among imbalanced data from different magnetic field strength.
Author Response
We express our gratitude to the reviewer for diligently and meticulously reviewing this article. Please see attachement to our anwsers to the reviewer remarks.

Reviewer 2 Report
In this data descriptor, the authors introduced a tumor and liver automatic segmentation (ATLAS) dataset to contrast-enhanced magnetic resonance imaging (CE-MRI) for hepatocellular carcinoma (HCC). It can be accepted after addressing several minor issues.
1. The authors should add some description about the analysis results in the abstract part.
2. The authors should give a comparation between their work and the previous reports to show the advantages of this study.
3. The figures should be supplemented with scale bars.
4. Please add more relevant references.
Author Response
The reviewer's diligent and meticulous assessment of this article is greatly appreciated. Please find attached our answers.

Reviewer 3 Report
This manuscript is dealing with the first dataset of on CE-MRI for HCC. The manuscript is well written and organized and contains results that could be interesting for some readers. According to my opinion, the list of references should be improved to include much more recent references.
Author Response
We express our gratitude to the reviewer for his contribution to this submission. Please find attached our answer.

Reviewer 4 Report
This study is to explore contrast-enhanced magnetic resonance imaging for hepatocellular carcinoma using A Tumor and Liver Automatic Segmentation (ATLAS) dataset. The topic is timely and the introduction is proper. Overall study processes are sound and valid. Findings are meaningful. The Following are some comments to improve the quality of the paper. - Add the study purpose in abstract and short findings. - Add one or two more keywords in the keywords section. - Add the study purpose in the introduction section. - Provide data validity on data collected for this study purpose. - If possible, provide some statistical results of train and test datasets. - If possible, provide some statistical test results between train and test datasets. - Add some more relevant references. - Double check all other minor concerns.
Author Response
We extend our thanks to the reviewer for their rigorous and detailed work on this article. Please find attached, our answers the reviewer's remarks.
